# Can Opioid-Free Anaesthesia Be Personalised? A Narrative Review

**DOI:** 10.3390/jpm13030500

**Published:** 2023-03-10

**Authors:** Jenna Goff, Morgan Hina, Nayaab Malik, Hannah McLardy, Finley Reilly, Matthew Robertson, Louis Ruddy, Faith Willox, Patrice Forget

**Affiliations:** 1School of Medicine, Medical Sciences and Nutrition, University of Aberdeen, Aberdeen AB25 2ZD, UK; 2Department of Anaesthesia, NHS Grampian, Aberdeen AB25 2ZD, UK; 3Pain AND Opioids after Surgery (PANDOS) European Society of Anaesthesia and Intensive Care (ESAIC) Research Group, 1000 Brussels, Belgium

**Keywords:** opioids, opioid-free anaesthesia, personalised anaesthesia

## Abstract

Background: A significant amount of evidence suggests that Opioid-Free Anaesthesia (OFA) may provide better outcomes for patients undergoing surgery, sparing patients who are particularly vulnerable to adverse side effects of opioids. However, to what extent personalizing OFA is feasible and beneficial has not been adequately described. Methods: We conducted a narrative literature review aiming to provide a comprehensive understanding of nociception and pain and its context within the field of OFA. Physiological (including monitoring), pharmacological, procedural (type of surgery), genetical and phenotypical (including patients’ conditions) were considered. Results: We did not find any monitoring robustly associated with improved outcomes. However, we found evidence supporting particular OFA indications, such as bariatric and cancer surgery. We found that vulnerable patients may benefit more from OFA, with an interesting field of research in patients suffering from vascular disease. We found a variety of techniques and medications making it impossible to consider OFA as a single technique. Our findings suggest that a vast field of research remains unexplored. In particular, a deeper understanding of nociception with an interest in its genetic and acquired contributors would be an excellent starting point paving the way for personalised OFA. Conclusion: Recent developments in OFA may present a more holistic approach, challenging the use of opioids. Understanding better nociception, given the variety of OFA techniques, may help to maximize their potential in different contexts and potential indications.

## 1. Introduction

The use of opioids in a medical capacity can be traced through thousands of years of human history [1]. In the 19th century, opioid synthesis and application was refined [2]. This is when opioid use started to become widespread across much of Europe, the United States of America, and the British Empire for both recreational and medicinal purposes [3]. The synthesis of fentanyl by Belgian physician Paul Janssen [4] further refined anaesthetic technique after its introduction in 1963. This enabled Gorge de Castro to create his theory of balanced, stress-free anaesthesia [5]. This technique was based on balancing anaesthetics with large doses of intravenous fentanyl (50 µg/kg) during surgical procedures. These methods were revolutionary due to their simplicity, safety (haemodynamic stability), minimal effect on haemostasis, and apparent comfort post-procedure—which had not yet been seen in the field [6].

In the 21st century, the concept of personalised anaesthesia came to prominence [7]. Personalised anaesthesia is based on the usage of population data studies by physicians to estimate dosages of anaesthetic agents. Anaesthetic agents have been found to produce over- and under-dosing, thereby increasing hospital stay length [8]. Thus, it is believed that personalised anaesthesia, through evaluating individual differences in anaesthetic sensitivity, could lead to better outcomes for patients [9]. We will discuss this idea to determine if such a practice is reasonably possible and desirable.

Our focus is on OFA, a recent area of research which potentially allows for decreased dependence on opioid class medications such as anaesthetic agents. The side effects of balanced anaesthesia are well studied with respect to their effects upon length of hospital stay and immediate recovery post-anaesthesia. These two factors make an alternative desirable and worthy of further research. Personalizing OFA will likely raise further research questions, especially if the monitoring of both patient and procedure-related factors are correlated with outcomes, therefore justifying resource investment.

## 2. Nociception Monitoring in OFA

Pain is described as an unpleasant sensory and emotional experience associated with actual or potential tissue damage [10]. Pain is distinct from nociception, which is the encoding of noxious stimuli within the CNS, developed as an evolutionary ability to allow organisms to associate potentially harmful stimuli with negative experiences [10]. Nociceptors, a type of sensory afferent neuron, convert noxious stimuli into electrical signals and can be classified as either Aδ or C fibres. Aδ fibres are highly myelinated, thereby allowing the rapid conduction (5–30 m/s) of well-localised nerve signals: ‘first pain’. C fibres remain unmyelinated and have a slower conduction (0.4–1.4 m/s): ‘second pain’ [11]. ‘Second pain’ is characterised by a poorly localised and long-lasting burning sensation. (See Figure 1 for more detail).

### 2.1. Opioid Pharmacology

Opioids act by binding to opioid receptors, of which there are three types (µ, δ and κ). All of these are GPCRs, coupled to inhibitory Gi/o and, subsequently, signal via the inhibition of AC to reduce intracellular cAMP levels. This results in the main effects of analgesia. Additionally, there are undesirable on-target side effects, including hyperalgesia, respiratory depression and immune suppression [12]. Opioid receptors are also known to signal βArr2 recruitment upon phosphorylation of the C-Terminal Domain of the receptor, mediated by GRK. This leads to a distinct set of signalling outcomes and clathrin-mediated endocytosis of the receptor. This process is known to re-sensitise receptors and is a major contributor to the phenomenon of tolerance (Figure 2) [13].

βArr2 signalling has been demonstrated to be responsible for many of the unwanted side effects of opioid use. Opioid analgesics biased in favour of inhibitory G protein recruitment may favour safer opioid analgesia in instances where total OFA is not feasible.

### 2.2. Single Parameter Scoring

#### 2.2.1. ANI

ANI (MDoloris Medical Systems, Lille, France) is a quantitative index scaling from 0–100. ANI utilises spectral analysis of HRV; specifically, the R-R distance between QRS complex per respiratory cycle in ECG traces [14]. ANI enables an enhanced intraoperative visualisation of the autonomic tone of the patient, whereby on the scale 0 represents a high parasympathetic modulation and 100 represents a low parasympathetic modulation (low and high stress responses, respectively) [15]. The target range of optimal analgesia is stated to be between 50–70, with values above and below this range suspected to be underdosing and overdosing with intraoperative opioid analgesics, respectively [16].

The NRS (Numerical Rating Scale) assesses the intensity of pain a patient is feeling from 0–10 and is commonly applied in all aspects of clinical practice due to its ease of use. The NRS score provided by patients when they arrive at PACU has been shown to correlate with an ANI (if <50 prior to extubation) and with an NRS > 3 with 86% specificity [17]. Therefore, ANI outputs in certain settings may be a potential predictor, even if imperfect, of the intensity of postoperative pain.

##### Limitations

ANI has been shown to be an ineffective measure if patients are either awake [18] or mildly sedated [19]. This may be because emotions and stress impact autonomic activity in conscious individuals.

Another major limitation of the ANI method is that many other sympathetic-altering medications are administered intraoperatively during anaesthesia (atropine or ephedrine, as well as different medications used in the context of OFA, including clonidine and dexmedetomidine), thereby confounding measures of sympathetic tone [20,21]. The autonomic tone of individuals also changes with age and between sexes, something which may also confound the obtained outputs [22].

So far, there is no consensus on the use of ANI in opioid-sparing analgesia. However, ANI has shown to be a more effective tool with certain agents (for example, remifentanil and propofol, when compared to sevoflurane and fentanyl) due to their differential effects on HRV. This highlights the need for further research on specific agent combinations [23].

#### 2.2.2. HFVI

HFVI (Mdoloris Medical Systems) is run on the same algorithm as ANI; HFVI and ANI are equivalent outputs. However, HFVI may be displayed concurrently with additional biometric information. In doing so, HFVI can form part of a multiparameter observation [24].

#### 2.2.3. Skin Conductance Measures

Skin conduction devices rely on rapid sympathetic changes through the skin based on the filling and excretion of sweat glands (namely palmar or plantar in neonates). This alters water permeability, thereby leading to micro-fluctuations in skin conductance detectable between two electrodes. The electrodes measure NFSC. If there are >0.2 fluctuations per second, there are links to severe postoperative pain as well as perioperative nociception [25,26].

In a recent article, skin conductance was employed to evaluate the favourability of OFA in elective thoracic surgery. Their finding was that skin conductance measurement demonstrated that OFA was associated with stable nociception during surgery (X).

Alas, skin conductance is restrained to allowing the visualisation of timepoints of heightened sympathetic activity. Consequently, its use in clinical practice is yet to be widely accepted.

#### 2.2.4. Pupillometric Assessment of Nociception

The control of the pupillary diameter is under strict sympathetic control. Alterations of the pupillary diameter in response to ambient light has been identified as a tool in monitoring intraoperative nociception [27]. The responsiveness to postoperative opioid treatment may be determined by the extent of pupillary changes in ambient light, as well as smaller changes observed prior to opioid analgesia linked to a decreased responsiveness to opioid treatment [28].

##### Limitations

It has been documented that the depth of anaesthesia can influence the degree of intraoperative pupillometric assessment [29]. One major limitation of this method of nociceptive monitoring is that intraoperative opioid use has been seen to have a direct effect on pupillary diameter and, therefore, confounds results [30]. Also of note is that all studies analysing pupillometric assessment of nociception have utilised small test groups and, as such, no clear outcomes have been identified regarding its implementation and clinical value.

Another potentially significant confounding factor is the hospital department of study. Studies conducted within ICUs have patient populations who present as being acutely unwell and may have trauma which comes before their admittance to hospital. This is contrasted with those in post-anaesthesia care units, where trauma mostly comes as a surgical consequence and is ongoing at the time of observation.

#### 2.2.5. NFR Threshold

The NFR threshold measures EMG activity in the bicep femoris muscle following the application of electrical stimuli to the ipsilateral sural nerve. Thus, the NFR threshold describes a polysynaptic spinal withdrawal reflex through the stimulation of Aδ fibres, something which precludes involuntary, perioperative movement. Considering even odds of 0.5, the NFR threshold demonstrates a predictive value of 0.63 at the 95% confidence interval. This predictive value is interpreted as being quite low. Furthermore, studies assessing the application of the NFR threshold are scarce; therefore, data are limited. Thus, currently, the value of using the NFR threshold during anaesthesia is questionable.

#### 2.2.6. BIS Monitoring

BIS is an EEG reading from electrodes applied to the patient’s forehead during surgery to characterise activity in their prefrontal cortex. Such EEG readings provide insight into the level of synaptic activity, which reflects the depth of anaesthesia. Consequently, appropriate titration of anaesthetic agents can be inferred. The output scale is from 0 (no electrical brain activity) to 100 (awake state). The range of values which allude to appropriate anaesthetic dosages are 40–60 during anaesthetic maintenance and 55–70 at the conclusion of the operation. However, despite large clinical trials employing BIS, there remains uncertainty around the definitions of perioperative “awareness”. Regardless, despite the low strength of evidence, it may be reasonable to use BIS to evaluate the risk of perioperative awareness, with positive implications for postoperative recovery and its management [31]. Ultimately, in the context of OFA, BIS may have a place in helping to personalise anaesthesia.

### 2.3. Two-Parameter Scoring

#### 2.3.1. SPI

The SPI is a two-parameter monitoring tool. These parameters are HBI and PPGA:SPI = 100 − (0.3 × HBI_norm + 0.7 × PPGA_norm)

When combined, these parameters give an indication of the counterpoise between sympathetic tone and parasympathetic tone [32]. The above equation returns a dimensionless value between 0 and 100, with lower scores indicating less nociception. An acceptable range of perioperative values is 20–50. When the SPI is employed, there is less postoperative opioid consumption as well as decreased patient arousal times [14].

#### 2.3.2. qCON and qNOX

The anaesthesia monitor Conox (Fresnius Kabi, Bad Homburg, Germany) records and processes EEG/EMG readings with the outputs of qCON and qNOX values [33]. qCON describes the depth of anaesthesia (consciousness), whereas qNOX reflects the level of analgesia (response to noxious stimuli). Consequently, these values can help predict involuntary, perioperative movement. The manufacturer describes a range of values between 0–99, based on a proprietary mathematical model, to indicate the likelihood of movement. Due to a lack of data, the clinical relevance of these scores is unresolved.

### 2.4. Multiparameter Scoring: NOL

NOL is recorded by a finger probe, then displayed on the Pain Monitoring Device (Medasense, Biometrics Ltd., Ramat Gan, Israel). This finger probe consists of an accelerometer, photoplethysmography, and measuring galvanic skin response and peripheral temperature. NOL values range between 0–100, with values >25 indicating nociception. Thus, a proposed acceptable range of values is 10–25 [34]. Whilst the NOL Index has been validated to be significantly superior to single-parameter monitoring and is an intuitive extrapolation from monitoring fewer parameters, there remains no established gold standard for perioperative nociception monitoring [15].

## 3. Relevant Drugs in Anaesthesia

GA is a drug-induced, reversible behavioural state that involves the maintenance of the physiologic systems (Table 1). Balanced GA consists of a triad of antinociception (analgesia), unconsciousness (amnesia) and immobility [35]. When inducing GA analgesics, sedatives, sympathomimetics and NMBA muscle relaxants are typically administered. Multimodal general anaesthesia builds on the principles of balanced general anaesthesia, which allows anaesthetists to develop an opioid-free regimen based on the patient’s needs.

Personalisation begins at the preoperative assessment. A regimen can be designed based on the patient’s history, choices, and present condition. Chronic pain, opioid use with prescription analgesics or illicit drug use may indicate opiate tolerance [36]. Some agent choices will also be influenced by the presence of comorbidities, while any history of psychiatric drug use, alcohol use, anxiety and pain catastrophising would all be factored into patient planning. An appropriate personalised approach would involve patient education, communicating mutual expectations and reducing anxiety through psychological preparation. Additionally, preoperative or adjuvant medications (such as benzodiazepines) serve as anxiolytics and mild sedatives with minimal respiratory depression [37].

When considering the multimodal strategy, anaesthetists choose the different anaesthetic combinations according to their primary (explicit) and secondary (implicit) effects (Table 1).

In practice, IV propofol, etomidate and ketamine are commonly used to induce GA. Lidocaine, midazolam and volatile anaesthetics often serve as adjuvants to the primary induction agent (Table 2) [38]. Balanced GA is based on the premise that combining drugs helps to reduce unwanted side effects by administering a lower dosage of each drug. These various agents act on different targets in the nociceptive system to control nociception intraoperatively and pain postoperatively. Opioids have traditionally been used almost exclusively in balanced GA. These multimodal practices aim to move towards ‘opioid-free’ or ‘opioid-sparing’ practice by utilising opioids after considering other agents first (Figure 3).

TIVA is becoming more favourable due to its inherent benefits and is inherently a personalised technique. It provides the opportunity to initiate and maintain anaesthesia with IV agents, while eliminating the need for inhalational agents. Environmentally harmful effects have been observed with inhaled anaesthetics [39], which also have side effects (such as PONV) which TIVA can avoid. Furthermore, TIVA benefits individuals with an increased risk of MH [40]. NICE recommends the concurrent use of BIS with TIVA, allowing the alteration of the infusion rate intraoperatively in response to a patient’s level of awareness [37]. This has been shown to reduce the risk of intraoperative awareness [41]. OFA covers different techniques and practices, including combining ketamine, lidocaine and magnesium sulphate infused through a single syringe infusion [42]. This combination, often in conjunction with an alpha-2 agonist and locoregional techniques, where indicated, has successfully produced OFA [43].

In the long-term, several benefits were observed, including decreased opioid consumption and tolerance, improved postoperative analgesia, reduced PONV and decreased pain profiles [43]. Combining the aforementioned medications into a single syringe is advantageous when multiple syringe drivers are unavailable—even if this is not approved in all countries. Additionally, the complexity of administering drugs is reduced when using a single infusion device, thereby reducing associated drug administration errors [42]. However, it may reduce the potential for a personalised approach.

The combination of certain agents has proved to have beneficial haemodynamic properties. When researching ketamine and magnesium sulphate for their potential use as adjuvants, both decreased haemodynamic variation. Furthermore, studies suggest this combination could be complementary regarding pain regulation and as established anaesthesia [43,44]. Hence, if haemodynamics and pain are variable, and variability is regulated, this may justify a personalised approach.

**Figure 3 jpm-13-00500-f003:**
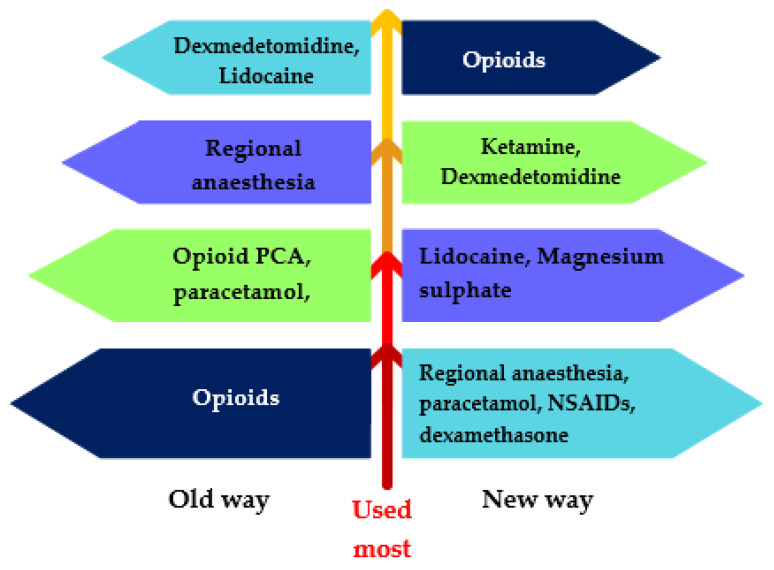
Multimodal pyramid. Adapted after [45].

## 4. OFA Personalised in Specific Specialties

In most cases, anaesthesia has always followed a ‘one-size-fits-all’ regime, which, over time, has been shown to have fewer desirable outcomes and decreased success rates in comparison to personalised therapy [46]. Personalised medicine is becoming more implementable as advancements in molecular technology, imaging modalities and genetic research have demonstrated growing evidence that there is inter-individual variability in a disease’s mechanism. The advancements in evidence and research have created increased motivation towards a personalised approach to medicine [47]. Personalised medicine allows medical professionals to predict a patient’s response to a disease and/or a particular therapy based on the patient’s unique ‘environmental exposure, physiological, molecular and behavioural levels’ [48]. By using different imaging modalities, monitoring devices and DNA sequencing, professionals can individualise a patient’s therapy based on their variation in characteristics which affect a disease’s development [48]. The need for advanced research in personalised OFA is prompted by current data, which indicate that certain patients are at a higher risk of opioid-related adverse effects based on their unique characteristics and contexts. These data include the increased risk of opioid tolerance and opioid-induced hyperalgesia in patients who already take opioids [49]. Currently, the lack of research means we cannot determine whether disregarding opioids for an individualised OFA regime is feasible for all patients [50]. In turn, this has led to a discussion about whether the personalisation of opioid-free anaesthesia is specifically desirable in specialities based on a particular patient’s inter-individual variability.

### 4.1. Bariatrics

As bariatric patients are at a significantly increased risk of perioperative opioid adverse effects, the speciality has seen a sizable volume of evidence on the use of OFA. These adverse effects include OIVI; more specifically, OIRD [51]. OIVI is one of the potentially life-threatening complications of opioid use [52]. This insinuates that it is advisable to avoid drugs which have an increased risk of respiratory depression (such as opioids) in patients who already have risk factors for respiratory depression before surgery; specifically, bariatric patients with obstructive sleep apnoea [53]. One systematic review and meta-analysis concluded that the incidence of OIRD can be reduced with the use of opioid-sparing strategies [54]. Opioid-sparing anaesthetic drugs include alpha-2 agonists (such as clonidine and dexmedetomidine), which reduce the anaesthetic requirements intraoperatively. This results in no significant respiratory depression, thereby indicating that both drugs could be a suitable and safer alternative to opioids in bariatric patients [55]. Ref. [56] supports the statement that dexmedetomidine may be safer than opioids in bariatric patients. In relation to the dexmedetomidine group of their small study, no patients required airway interventions, thereby contrasting with the placebo group. This suggests that dexmedetomidine improves patient safety as there is a significantly reduced risk of airway compromise. However, despite the risk of respiratory depression with opioid use, there are numerous studies which suggest that the risk of OIVI is dose-dependent [54,55]. This dose-dependent effect suggests that OIVI could potentially be prevented in certain bariatric patients if there was a better understanding of the pharmacology of certain opioids alongside improved knowledge on how to carefully titrate the opioid dose for each patient. The current evidence on this topic makes it difficult to determine if personalising anaesthesia, with low-dose opioids and opioid-sparing drugs, based on a patient’s risk factors is more desirable in comparison to completely disregarding opioids in all bariatric patients.

While the evidence reviewed here suggests that OFA within the bariatric speciality is possible, there is still a lack of convincing evidence to suggest that an OFA regime is easily implementable in every case of bariatric surgery. Additionally, despite the evidence suggesting the adoption of opioid-free regimes, we cannot confidently suggest the use of completely OFA due to the lack of evidence regarding the desirable combination of non-opioid alternatives, their combined adverse effects and their appropriate doses within patients—which may depend on both patient-related factors and practitioner expertise. This highlights the need for future research and data on OFA in a larger variety of patients and contexts. Alternatively, it may be possible to individualize a patient’s anaesthesia by combining low-dose opioids and opioid-sparing drugs by conducting preoperative screening based on genetics, preoperative questionnaires and imaging techniques on surgical patients to determine whether such patients are at elevated risk for adverse opioid complications. In turn, this aids the decision on which anaesthetic regimen is ‘the right therapy for the right patient, in the right dose and at the right time’ [9]. As each therapy is best suited to that individual patient’s needs, this would not only improve patient safety, it would also improve the quality of patient-centred care within anaesthesia.

### 4.2. Surgical Oncology

Patients undergoing cancer surgery often present specific challenges. Cancer is an inflammatory disease and, in some cases, is accompanied by a degree of immune suppression. These patients are treated with various modalities, many of which have an impact on the immune and nervous system. From a psychosocial point of view, they are at a high risk of experiencing persistent pain, including in survivors, which may impair their capacity to recover long-term. Therefore, it is important to limit any medications which may result in additional adverse side effects, such as PONV or hyperalgesia, which are associated with opioids.

There have been numerous studies conducted on the immunosuppressive effects of opioids. This causes concern that opioids may influence cancer recurrence and tumour growth. Opioids have been shown to adversely affect the activity of various immune cells, including NK cells [57]. NK cells are imperative in tumour surveillance by lysing tumour cells. It has been proposed that NK cells will play an important role in cancer treatment in the future [58]. It has also been hypothesised that opioids can have a direct effect on the progression of tumours and the recurrence of cancer. It has been shown in rodents that long-term exposure to high doses of opioids may limit tumour progression and that one-off or low doses of opioids, such as those used in cancer surgery, may encourage tumour growth [59]. Data have shown that high doses of intraoperative opioids may be associated with the recurrence rate of oesophageal squamous cell carcinoma, however not adenocarcinoma [60]. The dose of opioids administered during surgery has also been associated with the recurrence rate of other cancers, for example, prostate, oral and non-small-cell lung cancer; however, this has not been confirmed in RCTs using OFA [61]. The evidence in relation to the association of opioid dosage and cancer recurrence is not sufficient and requires further assessment. This is likely to require a tumour-specific approach, possibly personalised.

There have been many successful breast cancer surgeries undertaken using OFA, thereby indicating that OFA is a feasible option. Respiratory depression, pruritus or nausea and vomiting could be largely reduced by implementing OFA, with a limited number of patients requiring extra analgesia [62]. OFA eliminates the risk of any possible side effects which are associated with intraoperative opioids, thereby potentially allowing for a superior postoperative recovery. Another study trialled the use of OFA in combination with PECS block for breast cancer surgery [63]. The latter study compared opioid-free techniques (including the use of PECS block) with the traditional opioid techniques used in breast cancer surgery. It was found that incidence of PONV was significantly less in the non-opioid group compared to the opioid group. The non-opioid group also had lower analgesic requirements and lower VAS pain scores following surgery. There was no hypotension or bradycardia reported in the non-opioid group (which can be a concern when using some OFA medications, such as alpha-2 agonists). There were also no complications from the PECS block. Patient satisfaction was higher in the non-opioid group. Considering these trials, it can be argued that breast cancer surgery without the use of opioids is not only possible but superior for the patient and postoperative recovery.

Additionally, OFA has been trialled in other types of cancer surgeries. One study trialling OFA during lung cancer resection surgery found that patients receiving OFA spent, on average, one day less in hospital [64]. The patients receiving OFA also experienced significantly less pain at 1 h postoperatively compared with the opioid group. Pain (at hours 0 and 24) and postoperative analgesic requirements in this study were otherwise similar. This reiterates that OFA is, at least, equally effective—if not preferable. The observed variability in outcomes implies a need for the personalisation of OFA. For OFA to become a more common and reliable technique, more studies in several specialties need to be conducted.

### 4.3. Cardiac Surgery

In cardiac surgery, high-dose opioids (such as sufentanil and fentanyl) are traditionally considered favourable anaesthetics for their cardiac stability properties and their ability to dull surgical stress [65]. However, opioid-based drugs are also linked to both prolonged ventilatory intervention and a longer hospital stay. Thus, within the last three decades, a multimodal anaesthetic approach has been widely accepted to ‘fast-track’ cardiac patients. The ‘fast-track’ model consists of small doses of short-acting opioids, including remifentanil, alongside inhaled anaesthetics to be used as an anaesthetic agent. This method of anaesthesia allows for an early extubation following surgery and a shorter stay in the intensive care unit [66]. Therefore, if low-dose opioids are already widely considered to be effective as anaesthetic agents in cardiac surgery, why not attempt to completely discard opioids in this specialty? Cardiac surgery is one area of medicine where patients would highly benefit from personalised OFA, as these patients tend to suffer from multiple comorbidities and require additional care with the management of drugs compared to a young, healthy individual [67].

CVD is the principal cause of global mortality (it accounts for approximately 17.9 million deaths annually). The World Health Organisation states that ensuring patients receive the appropriate treatment is vital in avoiding early deaths [68]. One study found that 1 in 10 patients undergoing cardiothoracic surgery will continue taking opioids 90-days postoperatively, indicating opioid-based anaesthesia may increase chronic opioid use [69]. This statistic may be even higher in reality, as opioid-associated complications are likely to be underreported in cardiac surgical patients. Both opioid misuse and withdrawal can cause severe cardiovascular consequences, including ACS, with 315 out 100,000 ACS hospital admissions being due to drug overdoses [70]. Therefore, cardiac surgery patients may likely show promising outcomes with a personalised OFA protocol being implemented. Recent studies have indicated that non-opioids (such as dexmedetomidine) are useful as an induction agent. Patients undergoing cardiac surgery with an OFA protocol received a significantly lower amount of postoperative intravenous opioid (morphine) in comparison to patients with an opioid-based anaesthesia approach [71]. The patients in the OFA group also demonstrated less pain whilst coughing and presented with a lower frequency of atrial fibrillation. This suggests that not only does dexmedetomidine have an opioid-sparing effect, but it can also significantly improve post-surgical complications. Requiring less antihypertensive drugs, the use of dexmedetomidine combined with lidocaine is beneficial for providing haemodynamic stability [72]. These studies have highlighted that personalised OFA within the cardiac surgical specialty is feasible, desirable and should be considered by clinicians as an alternative to avoid unnecessary opioid consumption (amongst other advantages).

However, one major limitation of OFA that needs to be considered is the possibility that dexmedetomidine may induce severe bradycardia [73]. This may be largely dependent on the study context as a high-quality meta-analysis did not find this complication [74]. Further randomised trials need to be conducted to further investigate dexmedetomidine, and it is likely that drug titration, if not personalisation, may reduce the likelihood of side effects—if there are any.

## 5. Genetics and OFA

Having an understanding of genetic principles, such as the role an individual’s unique DNA plays in drug metabolism, is crucial in both pharmacology and anaesthetics. This phenomenon is known as ‘pharmacogenetics’ and was first described by Fredrich Vogel in 1959, after a 1947 scientific article found an imbalance of a certain protein to be associated with sickle cell anaemia, thereby suggesting that genetic alterations can, in fact, lead to illness [75]. Thus, researchers agree that personalised drugs directly targeting that specific mechanism, or error, may be more beneficial than a standard ‘one-size-fits-all’ drug. However, there is still a scarce amount of research available on the practicality of using genetic studies to tailor anaesthesia (particularly OFA) to patients, as some clinicians think that an appropriate titration of drugs to an individual’s needs is both sufficient and cost-effective. Even so, there are still numerous papers which explore the positive effects genetic studies may have on personalised medicine and opioid-based anaesthesia. Compared to such studies, we propose that the same theory could be applied to OFA, via incorporating a treatment regimen tailored to the individual according to their genetic profile. This tailored regime would aim to take a more patient-centred approach with improved safety. Pharmacogenetics is a specific term which describes the response which SNPs (changes found at a distinct point in a nucleotide sequence) have on drug therapy. In contrast, pharmacogenomics refers to the whole genome and the effect which several SNPs and mutations simultaneously have on drug metabolism [76]. Pharmacogenetics and pharmacogenomics combined allow researchers to study the influence that drugs have on gene expression while targeting certain sub-populations which may be better suited to a particular treatment [77].

Of the superfamily of enzymes, CYP1, CYP2 and CYP3 are the principal enzyme families which present with a role in drug metabolism, as these families operate approximately 80% of the body’s response to drugs [77]. The genes of these enzymes are susceptible to over 2000 mutations and have the capacity to disturb gene transcription and activity, ultimately affecting the CYP protein function. Any normal gene has two functional alleles to exhibit normal function. Researchers have categorised individuals into distinct phenotypes according to the activity of these CYP enzymes: from poor metabolisers (who fail to display any CYP activity due to the presence of two non-functional alleles leading to a non-functional enzyme) to ultrarapid metabolisers (who display extensive CYP activity due to an additional allele) [77]. The unique individualistic response to opioids used in anaesthesia can be ascertained via this method. CYP2D6 is an enzyme which converts opioid medications into an active form associated with an analgesic effect [78]. However, poor metabolisers will benefit from a limited effect only. Conversely, ultrarapid metabolisers encounter increased plasma morphine levels; therefore, they are more susceptible to opioid toxicity [77]. A patient who died of opioid toxicity was discovered to have an additional CYP2D6 functional allele in their genotype, thereby explaining their ultrarapid metabolism of opioids [78]. Thus, ultrarapid metabolisers could be considered for OFA, to ensure that they are not exposed to prodrugs metabolised by CYP2D6.

Numerous genetic variants have been discovered to exhibit an effect on the efficacy of anaesthetic agents and any side effects which they may produce. For example, mutations in the RYR1 can cause MH, a severe condition characteristic of abnormal calcium levels resulting in both hypoxia and hyperthermia. MH is elicited by volatile anaesthetics. Therefore, as a precaution, clinical theatres now store dantrolene because it decreases calcium release. Furthermore, patients who are found to have experienced MH receive a different anaesthetic agent—for example, TIVA, which is proven to be MH-safe—in any future surgeries. The use of dantrolene and TIVA have resulted in a reduced mortality of MH from 80% to 5%—a significant outcome due to clinical practice being amended and personalised as a result of genetic studies [77]. Considering this, we think the potential which personalised OFA with respect to genetic profiles could have on the anaesthetic field is worthy of further research.

## 6. Further Discussion and Perspectives

OFA can not only reduce anaesthesia’s reliance on opioids but can also be integrated into a more personalised approach. Therefore, the side effects of anaesthesia, in general, could be reduced. In this context, OFA provides an opportunity more than an obstacle.

The role of nociception monitoring devices remains unclear in OFA and warrants further investigation, particularly with respect to their ability to improve outcomes. Currently, such monitoring can guide clinicians interested in opioid reduction by providing information on the unhelpfulness of (high dose) opioids in many situations.

OFA personalisation involves not only medication choice, but the dose of those medications too. OFA provides the opportunity to tailor anaesthesia based on patient-centred goals, especially in the shared decision-making process (for example, early recovery and symptom control). Certain comorbidities and procedures may make patients particularly vulnerable to opioid-related side effects. For others, the effectiveness of opioids may be particularly unpredictable, such as patients already tolerant or dependent on opioids before surgery. This could inform potential specific indications for OFA, provided they are incorporated into a personalised approach.

While OFA has been highlighted as being transferable, it remains unclear how worthwhile the effort to set up and customise an OFA technique is. Personalization of anaesthesia may need to be prioritized first (over OFA), given the impact of techniques on postoperative outcomes.

Additionally, it also remains to be clarified to what extent some patients may experience, more than other patients, adverse events related to opioid-free techniques. Moreover, the context may influence implementability and safety, in particular education, sometimes more than evidence [79]. Bradycardia, for instance, has been observed when dexmedetomidine was delivered using an inappropriate protocol [80].

Finally, this work does not extensively discuss differences between total intravenous OFA and OFA combined with inhalation anaesthesia. While side effects of inhalation anaesthesia, like PONV, are well known, limited data, and no direct comparison, have been published in the specific context of OFA. In addition, we have not discussed OFA versus opioid-sparing techniques combined with intraoperative opioids, as this was not our main focus. However, we cannot exclude the possibility that the concepts discussed here may be valid in this context too.

## 7. Conclusions

The personalisation of OFA opens new research avenues. The variability of patient conditions should be better integrated in care, as they have implications for the effect of anaesthetic techniques. Perioperative nociception monitoring is an area of current research and there remains no gold standard in such monitoring. If the feasibility of OFA is established, then the appropriateness of OFA remains to be discussed, especially in teams lacking expertise. Personalised anaesthesia using a variety of approaches (including genotypes and phenotypes as well as shared decision-making) may help prioritise any future developments.

## Figures and Tables

**Figure 1 jpm-13-00500-f001:**
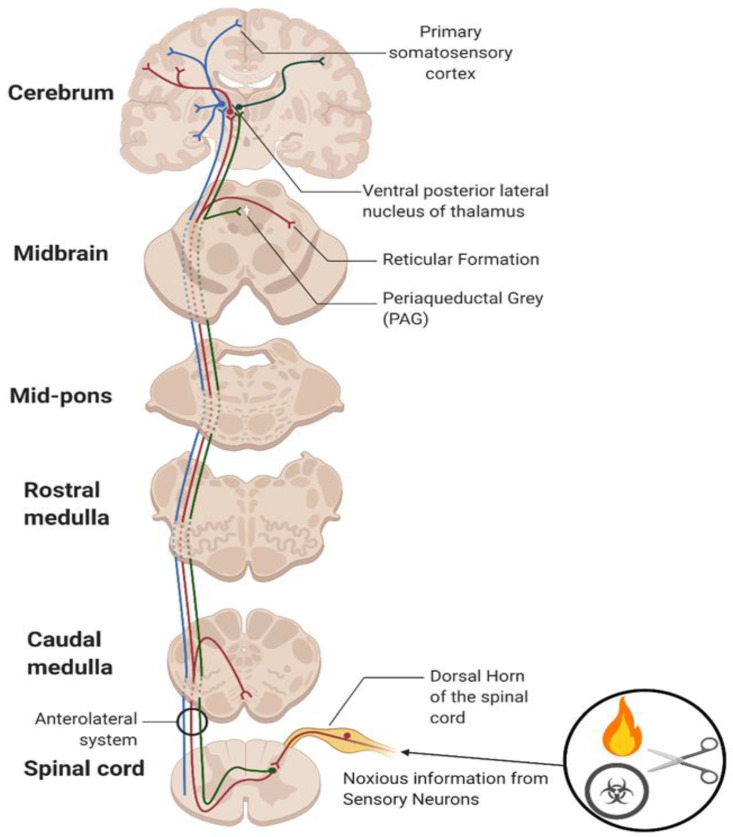
Illustration of the main nociceptive tracts. The Spinothalamic Tract (STT, blue) allows the localisation of pain. The Spinoreticular Tract (SRT, red) is more elusive and less understood; however, it may have roles in alterations of attention levels in pain responses due to it following the same tract as the STT. SRT also projects to areas of the Reticular Formation (RT) which is known to be associated with habitual pattern formation. The Spinomesencephalic Tract (SMT, green) is responsible for the modulation and inhibition of pain. Fibres in the SMT also project to areas of the Periaqueductal Grey (PAG), which is known to be responsible for the control of pain through the activation of enkephalin releasing neurons.

**Figure 2 jpm-13-00500-f002:**
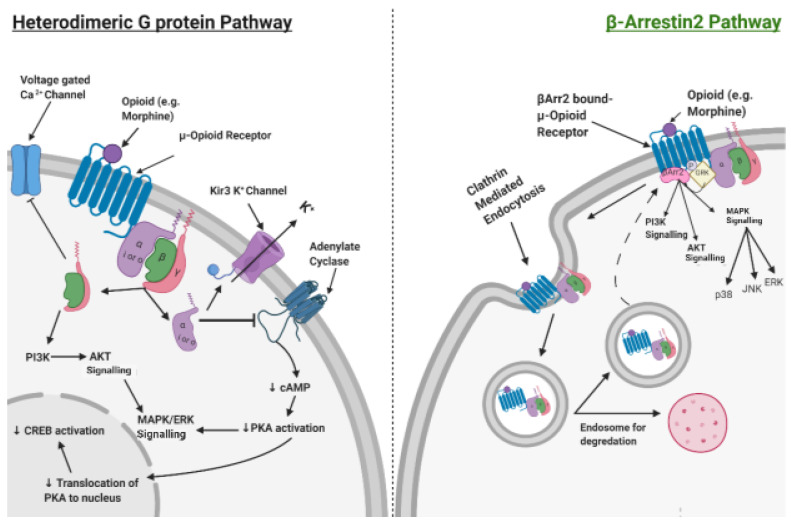
Cellular mechanisms of both Heterodimeric G Protein opioid signalling and Beta-Arrestin 2 (βArr2) mediated signalling pathways. Note that the βArr2 pathway is implicated in the phenomenon of tolerance.

**Table 1 jpm-13-00500-t001:** Multimodal general anaesthesia: primary and secondary effects of agents.

Role in Anaesthesia	Primary Effect of Agent	Secondary Effect of Agent
Antinociception (analgesia)	Ketamine (NMDA antagonist) Remifentanil (opioid agonist) Dexmedetomidine and clonidine (alpha-2 agonists) Magnesium (NMDA antagonist) Lidocaine (anti-inflammatory; sodium channel blockade) NSAIDs (anti-inflammatory)	Propofol (GABA agonist) Sevoflurane (GABA agonist and other targets)
Unconsciousness (amnesia)	Propofol (induction and maintenance) (IV) Sevoflurane (inhaled)	Ketamine (NMDA antagonist) Remifentanil (opioid agonist) Dexmedetomidine (alpha-2 adrenergic agonist) Magnesium (NMDA antagonist)
Immobility (muscle relaxant)	Cisatracurium (induction and maintenance) Rocuronium (induction and maintenance) Succinylcholine (induction)	Magnesium (smooth muscle relaxant) Propofol (central muscle relaxation) Sevoflurane (central muscle relaxation)

Examples of GA agents used to maintain balanced GA. **GABA**, gamma-aminobutyric acid; **NMDA**, N-methyl-D-aspartate; **NSAIDs**, nonsteroidal anti-inflammatory drugs. Based on reference [35].

**Table 2 jpm-13-00500-t002:** Antinociceptive agents: mechanisms of antinociception, sedation and unconsciousness.

Agent	Mechanism and Function	Benefits	Risks and Contraindications
**Opioids (fentanyl, morphine)** **(antinociception)**	μ-receptor agonist. Binds to opioid receptors, targeting peripheral inputs and inhibiting nociceptive stimulation	During intubation laryngoscopy: airway reflex suppressed, reduced stress response, (indicated by tachycardia and hypertension). During induction agent administration: Less pain from inducing agent, and supplements sedation, thus reducing inducing agent dose	Increased risk of respiratory depression, which can be dose-dependent, possible apnoea, PONV
**Propofol**	Induces anaesthesia by enhancing GABA-A transmission	Antipruritic, antiemetic and reduced post-op sedation	Can cause twitching, hiccups, and pain on infusion. Can reduce cerebral perfusion
**Esmolol**	Short-acting, cardio-selective β1-adrenergic receptor antagonist	Alternative to opioids for maintaining haemodynamic and BIS stability during GA. Safe for use with pulmonary disease, paediatrics	No analgesic effect when given alone. Use is risky in diabetics and peripheral circulatory disorders
**Dexmedetomidine, clonidine**	Alpha-2 adrenergic receptor agonists, thus inhibiting sympathetic response. Has a role in sedation and acts as an adjuvant to neuraxial anaesthesia	Offers sedation without high risk of respiratory depression. Has a wide safety margin	Severe bradycardia, asystole in some studies but not in meta-analysis
**NSAIDs**	Analgesic due to inhibition of COX enzymes and inflammatory mediators	Cheap, accessible and patient can self-administer	Risk of GI irritation, impaired coagulation and renal insufficiency
**Paracetamol**	Analgesic and antipyretic effect due to inhibiting prostaglandin synthesis and possible central effects	Usually well-tolerated, especially in bariatric populations	Metabolism adversely affected by liver dysfunction
**IV Lidocaine**	Blocks neutrophil priming, impairing the ability of cells to amplify inflammatory response and, consequently, nociceptive signalling. Antiarrhythmic and analgesic effect created by blocking Na+ channels, thus preventing nerve transmission	Wide safety margin, and usually well-tolerated regardless of administration route. IV lidocaine infusions generally accompanied by feeling of sedation	Caution advised if an arrythmia is present that is not being treated with lidocaine, and toxicity can cause systemic neurological symptoms
**Ketamine**	Homeostatic and analgesic effect as an NMDA antagonist	Rapid onset with low risk of respiratory or haemodynamic depression	Patients can experience dysphoria and unpleasant hallucinations
**Magnesium sulphate**	Homeostatic and antiarrhythmic acting on voltage dependent Ca^2+^ channels. Non-competitive NMDA antagonist. Antinociceptive mechanism similar to ketamine	Acts as an adjuvant to other analgesics, reducing required dosages, and preventing sensitization due to nociceptive stimulation	Risk of toxicity and exaggerated hypotension

COX, cyclooxygenase; NMDA, *N*-methyl-D-aspartate; NSAIDs, nonsteroidal anti-inflammatory drugs; PONV, Postoperative nausea and vomiting; IV, intravenous; GI, gastrointestinal; GABA, gamma-aminobutyric acid.

## Data Availability

Not applicable.

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
