# Peer review of "Can Opioid-Free Anaesthesia Be Personalised? A Narrative Review"

_jpm, 2023, doi:10.3390/jpm13030500_

Round 1
Reviewer 1 Report
The authors conducted a narrative literature review aiming to provide a comprehensive understanding of nociception and pain and its context within the field of Opioid-Free Anaesthesia.
The proposed subject is relevant and important, and the information provided may be considered an important subject.
I have no principal issues with the study that is very well-designed and built-in clear accordance with the best principles of scientific writing.
I think it is an article of merit and interest in this field that should be published at it is presented.
Author Response
Dear Reviewer,
Thank you kindly for the positive comments on our work, we hope to see publication soon.
Kind regards,
Matt
Reviewer 2 Report
Dear authors
I find your article interesting and well organized.
Please include some perspectives of complications of the opioid free anesthesia and potential for chronic pain and inadequacy of anesthesia.
Author Response
Dear Reviewer,
Your suggested additions have been included with the manuscript, thank you for your valuable contribution.
Kind regards,
Matt
Reviewer 3 Report
The paper is very interesting and raises important questions from the point of view of practice. This form of research methodology is justified due to the breadth of the problem and the lack of possibility of accurate comparisons between assessment methods.
The work is highly valuable and based on the current state of knowledge.
Please analyze the appropriateness of using capital letters in some places
Please consider including the following article, which relates to measurements using the Skin Conductance Algesimeter .. Opioid-Free Anaesthesia Effectiveness in Thoracic Surgery-Objective Measurement with a Skin Conductance Algesimeter: A Randomized Controlled Trial
Author Response
Dear Reviewer, many thanks for your guidance on expanding the use of skin conductance during anaesthesia. We have amended our draft to satisfy this input.
Kind Regards,
Matt Robertson
Reviewer 4 Report
The title seems misleading. The whole review is about introduction of OFA and the need for paradigm shift from the world of opioids.
When we talk of personalisation, we start from the assumption that the technique is effective and can be adapted easily.
Moreover, the concept of OFA in which context? With conventional anaesthesia where opoids are replaced by other agents or another shift from inhalation anaesthesia to TIVA?. This distinction is lacking from the review. Also the concept of opioid sparing where as OFA needs to be discussed. There are a few controversies around OFA which should have been made clearer by the review, unfortunately they have nit been addressed.
Author Response
We thank the reviewer for the comment although we respectfully disagree that the work is not about personalisation. Personalised medicine has more than one definition. It's not the purpose of the work to discuss these different definitions, but, if we use the NHS's one, "Treatments based on what would be most effective for you and experience fewer, if any, side effects", we believe we have discussed how to adapt OFA techniques based on maximising effectiveness and minimising side effects in different populations and contexts. We agree with the reviewer that it remains unclear whether this is easy or not, and have carefully reviewed our conclusion, aligned on that. It's also reflected in the title by keeping the question mark, implying we don't pretend to come with a definitive conclusion. Finally, the title was chosen in response to an editor demand. Of course, if the editor asks for a change in the title, we will agree.
We agree with the reviewer that we haven't discussed differences between total intravenous OFA and OFA combined with inhalation anaesthesia, and OFA versus opioid-sparing techniques combined with intraoperative opioids. We have now highlighted that in the conclusion, but think that expanding more would be mostly speculative and out of the scope of the work. Here is the text we have added: "Finally, this work does not extensively discuss differences between total intravenous OFA and OFA combined with inhalation anaesthesia. While side effects of inhalation anaesthesia, like PONV, are well known, limited data, and no direct comparison, has been published in the specific context of OFA. Also, we haven't discussed OFA versus opioid-sparing techniques combined with intraoperative opioids, as this was not our main focus. However, we cannot exclude the possibility that the concepts discussed here may be valid in this context too." Thank you very much for taking the time to read and review, the team and I truly appreciate it. Kind regards, Matt RobertsonRound 2
Reviewer 4 Report
The addition of the paragraph clarifies a lot. A small correction may be made in table 1. Muscle relaxants are nit used for induction or maintenance of Anaesthesia. The terms used may mislead the readers. It should be clarified that they are used during induction/ maintenance for muscle paralysis only without any analgesic or amnesiac effects. In some parts of the world, induction and maintenance are terms solely used for anaesthesia and muscle paralysis is entirely a separate entity.